

# Inhomogeneous precursor characteristics of rock with prefabricated cracks before fracture and its implication for earthquake monitoring

Andong Xu[1], Yonghong Zhao[1], Muhammad Irfan Ehsan[1], Jiaying Yang[1], Qi Zhang[1], Ru Liu[1]

[1]School of Earth and Space Sciences, Peking University, Beijing, 100871, China

*Correspondence to*: Andong Xu(andongxu@pku.edu.cn)

**Abstract.** Earthquake precursor and earthquake monitoring are always important in the earthquake research field, even if there is still debatable about the existence of earthquake precursor. Laboratory rupture experiment is a useful technique to simulate and make an insight into the complex mechanisms of earthquakes. Five marble samples with prefabricated cracks are used for uniaxial loading experiments to investigate whether there is a precursory signal before rock fracture, and to

simulate the rupture process of strike-slip fault. The existence of precursory signal is confirmed by the coefficient of variation (CV) results, from which we can see two patterns which are known as seismicity acceleration and quiescence before an earthquake. Moreover, these CV findings are applied to determine the locations of large deformation sampling points on the rock surface at different loading stages. Similar results are obtained when we consider actual seismicity at the northern end of the San Andreas Fault in California, which provides a crucial evidence to prove the existence and role of

precursor characteristics. In this case, three kinds of seismic monitoring models are designed to find out how to monitor these precursor characteristics more effectively.

## 1 Introduction

The issues of earthquake such as initiation, growing and monitoring are difficult but attractive. Considerable efforts have been done to understand earthquake source mechanisms (Goff et al., 1987; Frohlich and Apperson, 1992; Frohlich, 2001;

Kagan, 1991, 2005, 2013; Aldamegh et al., 2009; Butler, 2019). It is now generally believed that earthquakes are caused by a sudden release of accumulated energy, which induced a sudden failure of intact rock or sudden stick-slip motions on pre-existing faults. The essential factors which affect these sudden stick-slip motions depend on fault properties, but it is extremely difficult to directly measure these properties such as friction strength and stress state. Laboratory rock experiments are a useful approach to make insights into rock and fault properties, including rate-and-state friction (Dieterich, 1979; Ruina,

1983; Rubin, 2008), and deformation under different conditions, such as in torsion or low temperature (Paterson and Olgaard, 2000; Beeler, 2007). Some of the other experiments which are known as laboratory earthquakes provide deep understanding of dynamic rupture process including supershear (Xia et al., 2004, 2005; Kammer et al., 2018) and fracture energy (Lockner et al., 1991; Kammer and McLaskey, 2019). However, it is still an unsolved problem as Kammer and McLaskey (2019) said





that how these laboratory observations should be scaled to the size and rates of naturally occurring earthquake fault ruptures. We try to make a different type of analysis to link the laboratory observations with natural seismicity by comparing their similar characteristics.

More and more precursors of rock fracture have been observed under the progress of rock experiment in the laboratory. Brace et al. (1966) explored that the dense igneous rocks increase in volume before fracture. Under differential stress, rocks dilate before failure, which is caused by the development of new cracks within the rock. These observations led Nur (1972) to suggest that, under stress, the ratio of Vp/Vs (Where Vp is the seismic P-wave velocity and Vs is the seismic S-wave velocity) should decrease if the rock becomes dilatant, and then increase again if water flows into the cracks from the surrounding regions. When a rock is stressed to failure, cracking on a microscopic scale occurs. These microcracks activity is known as acoustic emission and considered by many to be simply a scale model of seismicity in the earth. There is a strong correlation between the amount of nonelastic strain and the number of acoustic emission events (Scholz, 1968). Some researchers find that as the rock approaches fracture the acoustic emission rate increases (Scholz, 1968; Lockner and Byerlee, 1977), while others have discovered a decrease just before failure (Brady, 1975; Kahir, 1977). Acoustic emission has been used to predict rock bursts in deep mines in the late 1930 (Obert, 1977). In order to analyze the rock properties so that for understanding the natural dynamic rupture processes, characteristics and deformation of rock fracture have been observed with development of experimental technique, including the influence of crack size on the fracture behavior (Harlin and Willis, 1990), scaling and universality in rock fracture (Davidsen et al., 2007) and triggering processes in rock fracture (Davidsen et al., 2017). As the research of these deformation characteristics is more and more in-depth, it is then a reasonable step to consider the possible precursor of these deformation characteristics before rock fracture.

We introduce an attribute statistic which is called as the coefficient of variation (CV) to quantify the deformation characteristics of rock fracture, to find the potential precursor to make a good coupling between laboratory experiments with natural earthquakes. We use five marble rock samples with prefabricated cracks to simulate the actual strike-slip fault such as the northern end of the San Andreas Fault (SAF), and to analyze the process of dynamic rupture during loading with digital speckle correlation method (DSCM) (Peters and Ranson, 1982; Yamaguchi, 1981; Ma et al., 2004). By quantifying the deformation of rock fracture, the precursor characteristics are been identified. These features are used to determine the position of the sampling points with relatively large deformation and detect their changes with the increase of load. We then study the distribution of epicenters in the seismic catalogue near the northern end of the SAF and try to compare the experimental results with it in order to investigate the common features between them. Finally, considering the actual situation, we design different seismic monitoring models and compare their monitoring effects on precursor, hoping to provide some guidance for the earthquake monitoring work.



## 2 Methods

### 2.1 Digital speckle correlation method (DSCM)

Digital speckle correlation method (DSCM) was proposed by Peters and Ranson (1982) and Yamaguchi (1981) in the early 1980s, respectively. The basic governing phenomena of DSCM is to calculate the correlation coefficient between the source image and the target image, as shown in Fig. 1. Take the speckle pattern f around the sampling point P in the source image and search on the target image by a certain correlation function (Ma et al., 2004) to find the best matching speckle pattern g. The corresponding P' in g is the sampling point p after deformation. The difference in pixel coordinates between these two points ((u, v)) is the displacement of sampling point p before and after deformation and its derivative represents the strain.

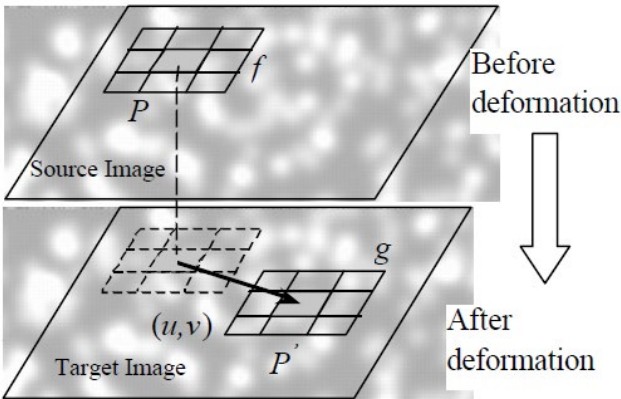

**Figure 1.** Schematic diagram of digital speckle correlation method (Ma et al., 2004)

DSCM extracts the displacement and strain information from random speckle signals produced by artificial or natural texture. Dynamic measurement can be achieved by high-speed video recording or high-speed photography system as DSCM is the direct solving process of two recorded images. We used artificial speckle and photography system (about 3.4 frames per second) to record the deformation images of the marble samples with prefabricated cracks during loading. By recording the images under different loads, the surface displacement and strain of the samples are worked out by DSCM so as to help finding the potential precursor of rock fracture.

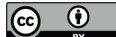


## 2.2 Experiments with uniaxial loading

Five marble samples with prefabricated cracks were used in the experiments in order to discover and analyze the common rather than unique precursor characteristics of their fractures under loading. The five marble samples are all prepared as shown in Fig. 2a and the deep black lines on the sample in this figure are precast cracks that are used for simulating the

actual strike-slip faults such as northern end of the SAF. We analyzed the dynamic rupture process during loading by using DSCM.

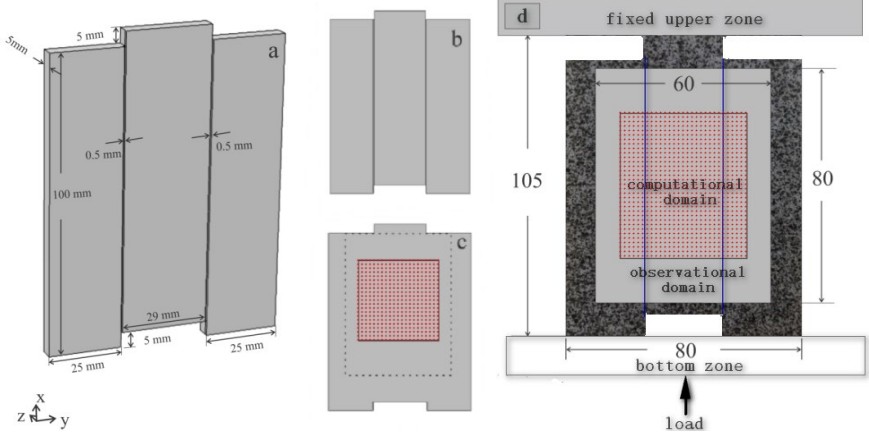

**Figure 2.** Schematic diagram of the sample and experimental device

Three-dimensional shape of the sample. The deep black lines indicate the prefabricated cracks. L is the length of the upper surface while

W is the width of it. The directions of 3D coordinates are shown in the lower left corner. (b) Two-dimensional shape of the sample. (c) Two-dimensional shape of the sample. The deep blue lines indicate the precast cracks, the area enclosed by the dash-line rectangle is observational domain and the solid-line rectangle is computational domain. The red dots spaced five pixels apart from each other are the sampling points in computational domain. (d) Experimental loading mode, observational and computational domain of the sample. There are random artificial speckle signals on the sample. The arrow indicates the direction of loading. The gray zone represents observational

domain and the area enclosed by the rectangle is computational domain.

The size indicated in Fig. 2a is ideal, and there may be some deviation in the actual production of the samples. We sprayed speckles on the surface of these five marble samples to construct grayscale characteristics which can be used for DSCM. Before the experiments, we measured the length and width of the upper surface of each sample since load would act on it (i.e.

L and W in Fig. 2a). The detailed data of the five samples were listed in Table. 1.



**Table 1.** The detailed data of the five samples

| Experimental data \ Samples | Sample 1 | Sample 2 | Sample 3 | Sample 4 | Sample 5 |
|---|---|---|---|---|---|
| L (unit: mm) | 29.54 | 29.42 | 29.50 | 30.00 | 29.20 |
| W (unit: mm) | 5.10 | 5.12 | 5.08 | 4.86 | 5.50 |
| Fracture load (unit: T) | 1.60 | 1.27 | 1.40 | 1.05 | 0.92 |
| Number of photos | 713 | 164 | 613 | 420 | 1177 |

All of the experiments are performed in the same way on a uniaxial loading apparatus whose upper zone could be fixed as shown in Fig. 2d. The direction of loading was also shown in Fig. 2d and the rate of load increase was artificially controlled. A photography system (about 3.4 frames per second) is used to film the entire process from initiation to destruction of the

samples during loading. The number of photos we got for each sample during loading was listed in Table. 1. We calculated all the photos using DSCM in the case of the selected observation and computation domains shown in Fig. 2d.

**2.3 Coefficient of variation (CV)**

The coefficient of variation (CV), defined by formula (1), is a statistical relationship which is used to describe the dispersion degree of a set of data ( $X_1, X_2, ..., X_n$ ).

$$CV = \frac{\sigma}{\mu} \tag{1}$$

Where $\sigma$ represents the standard deviation and $\mu$ is the mean value of the data set. The computation of $\sigma$ is described in Eq. (2)

$$\sigma = \sqrt{\frac{\sum_{i=1}^{n}(X_i - \mu)^2}{n-1}} \tag{2}$$

and the mathematical expression of $\mu$ is given in Eq. (3).

$$\mu = \frac{\sum_{i=1}^{n} X_i}{n} \tag{3}$$





Kagan and Jackson (1991) used this statistic to describe the clustering of earthquake inter-occurrence time for investigating the characteristics of long-term earthquakes. Here, we applied the CV to describe the precursory characteristics of failure of the samples with prefabricated cracks, since the deformation degree of each part of the samples is intuitively different with the increase of load due to the existence of the precast cracks. In other words, we wanted to use this statistic to find out whether there is a significant signal before the samples fracture since the dispersion degree of the data fluctuates with loading. We have chosen the image of each sample in the initial state (i.e. without load) as the source image, and the third, fifth, seventh and so on till the destruction image of each sample as the target images, which were all taken by the photography system. Various data of the samples can be obtained with DSCM, including the displacement and strain of each sampling point in the computational region. Therefore, selecting the proper data to calculate the CV is the next crucial step.

Actually, the data we have recorded in real life, such as GPS data, crustal stress data and other data, are all compared with a certain state rather than the initial state because we cannot know and record the initial state of a natural area. Thus, we proposed a so-called increment method to calculate our data so as to be consistent with the actual situation. Firstly, we obtained the displacement and strain of every moment during loading by selecting the initial state image as source image and the later state images as target images. Then, the differential displacement and strain of each sampling point in the computational domain were acquired by subtracting the results of the previous moment from the results of the later moment, which is what we call increment. The displacement and strain of each sampling point obtained by this method constituted what we call differential displacement field and differential strain field. It is worth noting that we focus on the dispersion degree rather than the positive or negative characteristics of the data, so we calculated the CV after taking the absolute value of the increment. Except for the linear and shear strain what we can obtain from DSCM directly, we have also considered the maximum and minimum principal strains. Relationship for calculating these two strains are shown below.

$$\varepsilon_{max} = \frac{\varepsilon_x + \varepsilon_y}{2} + \sqrt{\left(\frac{\varepsilon_x - \varepsilon_y}{2}\right)^2 + \frac{\gamma_{xy}^2}{4}}$$

$$\varepsilon_{min} = \frac{\varepsilon_x + \varepsilon_y}{2} - \sqrt{\left(\frac{\varepsilon_x - \varepsilon_y}{2}\right)^2 + \frac{\gamma_{xy}^2}{4}} \qquad (4)$$

$$\gamma_{xy} = \frac{\partial u}{\partial y} + \frac{\partial v}{\partial x} = 2\varepsilon_{xy}$$


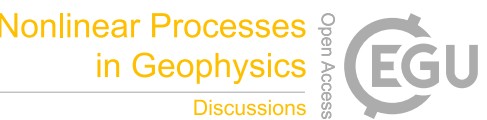

Where $\varepsilon_x$ and $^{u}$ are linear strain and displacement in parallel direction of load; $\varepsilon_y$ and $^{v}$ are linear strain and displacement

in vertical direction of load; $\varepsilon_{xy}$ and $\gamma_{xy}$ are shear strain in rock mechanics and engineering, respectively. The calculation

step of the corresponding differential maximum and minimum principal strain field were consistent with increment method.

The CV of all data above are calculated by using equations (1, 2, 3), and their images with increasing load are shown in Fig.

3. Here, we put the CV images of sample 4 for analyzing since its characteristics are clear. The similar images of the other

samples are in Appendix A.

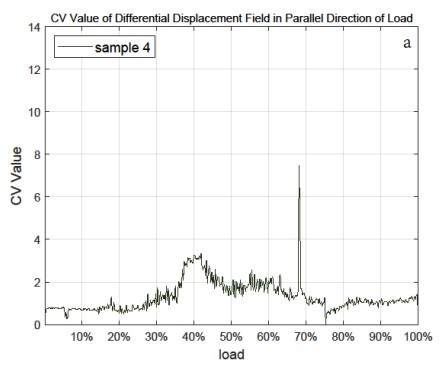
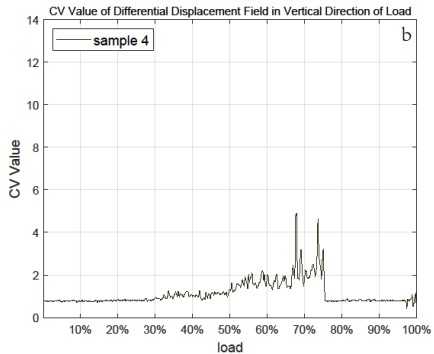


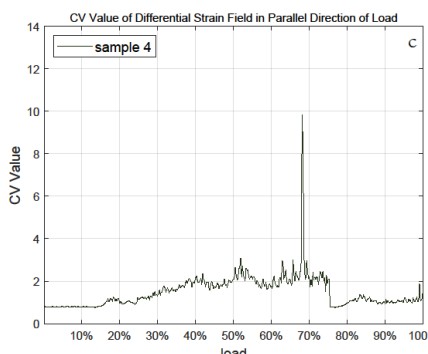
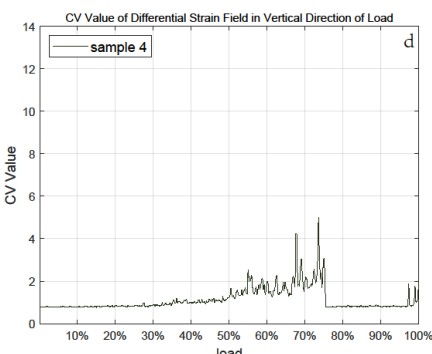





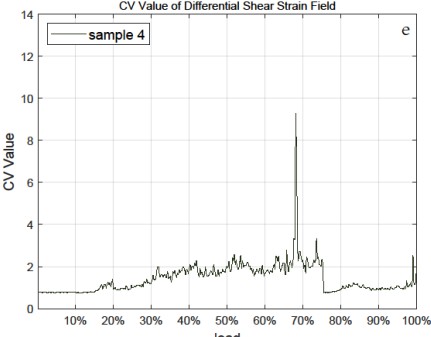
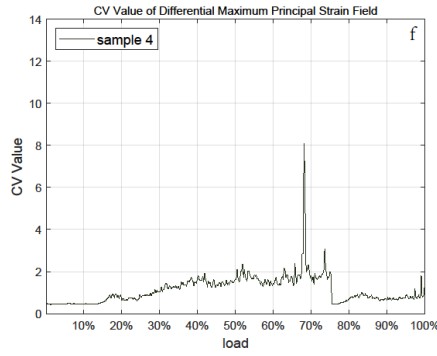

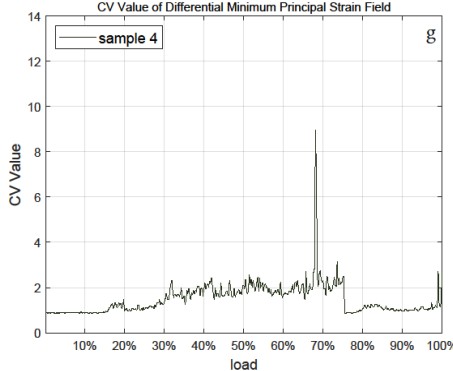

**Figure 3.** CV images of sample 4

(a) The image of CV changing with load, which is obtained from the differential displacement field in parallel direction of load. (b)The image of CV changing with load, which is obtained from the differential displacement field in vertical direction of load. (c) The image of CV changing with load, which is obtained from the differential strain field in parallel direction of load. (d)The image of CV changing with load, which is obtained from the differential strain field in vertical direction of load. (e) The image of CV changing with load, which is obtained from the differential shear strain field. (f) The image of CV changing with load, which is obtained from the differential maximum principal field. (g) The image of CV changing with load, which is obtained from the differential minimum principal strain field.

As can be seen from these images, the CV calculated by different physical quantities fluctuate with load. Some of them show a significant jump before rock fracture (100% loading stage), which is what we call the precursory characteristic of fracture. This kind of precursor can appear when the CV of proper physical quantities are monitored and calculated. As for

the experimental results, we believe that the CV results obtained by calculating the differential strain field are better than those got from the differential displacement field. Because the displacement is actually very sensitive to loading, the displacement of each sampling point is relatively large at the laboratory scale. In this case, the variation of CV is not so obvious, which can be seen in the CV images of multiple samples (Appendix A). In contrast, the strain field can often reflect

the concentration of deformation, so it is useful for extracting the dispersion characteristics of the data and such precursory signals. Furthermore, the maximum principal strain is generated by the maximum principal stress, and the CV calculated by this strain has obvious precursor signals, such as the significant jumps during 60 to 80% and near 100% loading stage shown in Fig. 3f. Considering above two factors, we take the differential maximum principal strain as the monitoring signal of the earthquake monitoring models.

## 2.4 Compare the experimental results with the natural seismicity

The differential maximum principal strain is also used to distinguish the sampling points with large deformation, which contribute to the CV jump. It is not difficult to notice that the CV reaches 8 at around 60-80% loading stage in Fig. 3f, so we have taken 8 as a judgement condition (which is also called threshold) to find these large deformation points described above for sample 4. Here we still take the results of sample 4 for analysis. At every 10% loading stage, the differential maximum

principal strain of each sampling points is compared with the average differential maximum principal strain of all sampling points. If the differential maximum principal strain of a sampling point is 8 or more times larger than the average value of all sampling points, we will mark the position of this sampling point on the surface of sample 4 and try to display all the qualified sampling points at the same stage. It should be mentioned that the threshold of different samples is not the same according to the CV of samples. Therefore, we take 4 as a threshold for sample 1, 3 for sample 2 and 3, and 4 for sample 5.

The results of sample 4 at each stage are shown in Fig. 4 and the other samples' results are shown in Appendix B.








**Figure 4.** The position of the sampling points with large differential maximum principal strain for sample 4

The blue lines indicate the prefabricated cracks. Each figure shows the observational area of sample 4 in the experiments. The area enclosed by the white rectangle is the calculation domain and its size is constant in different loading stages. The red points represent the

sampling points with large differential maximum principal strain that satisfy the judgment condition as the load increases.

(a) The result of the 10% loading stage. (b) The result of the 20% loading stage. (c) The result of the 30% loading stage. (d) The result of the 40% loading stage. (e) The result of the 50% loading stage. (f) The result of the 60% loading stage. (g) The result of the 70% loading stage. (h) The result of the 80% loading stage. (i) The result of the 90% loading stage. (j) The result of the 100% loading stage.

Fig. 4c shows that some sampling points with large differential maximum principal strain that satisfy the judgement condition began to appear at 30% loading stage, which corresponds to the phenomenon that the CV starts to rise in Fig. 3f. Then the position of such sampling points changes with the increase of load, and the deformation becomes larger and larger. According to the Fig. 3f, the CV reaches the maximum level at about 70% fracture load, and then enters the quiet period until sample 4 is broken when the load is 100%. When sample 4 approaches fracture, the sampling points with large

deformation are concentrated near the precast crack. The consequences of the other samples also show this concentration phenomenon, which leads us to have an interest in investigating the location of earthquakes near strike-slip faults. Exactly, we want to know how the location of small earthquakes near a strike-slip fault changes over time and where the major earthquake occurs during an earthquake cycle.

    The SAF is a strike-slip fault formed by the relative motion of the Pacific and North American plates. It is a seismically

active area with a rich seismic catalogue. There are many studies for this area by using the seismic catalogue (Gutenberg and Richter, 1945; Thurber et al., 2004; Barbot et al., 2012). We also focus on the seismicity of the SAF because the prefabricated cracks in the sample are used to simulate the deformation characteristics of the strike-slip fault and its surrounding area when the load increases over time. In order to eliminate the influence of other faults, we actually chose the area enclosed by the blue rectangle in Fig. 5 as the research area. The longitude and latitude of these four points in the blue

rectangle are A (121.4000°W, 36.4949°N), B (120.5000°W, 35.6174°N), C (120.1500°W, 35.9764°N) and D (121.0500°W, 36.8539°N). After selecting the study area, it is necessary to determine starting and ending time of the seismic catalogue. Because the experiments are about the deformation and fracture precursors of prefabricated-cracks samples in a complete



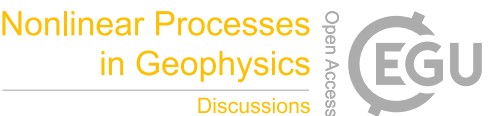

loading period, a complete seismic cycle is also needed at the time of selection of earthquake catalogue. We have taken the

occurrence time of the Parkfield Mw 6.0 earthquake (the epicenter is 120.3660°W, 35.8182°N) that happened in the study

area on September 28, 2004 as the termination time. Then, the third month after the last earthquake (magnitude ≥ 5) occurred

in the study area is chosen as the starting time in order to eliminate the influence of aftershocks. Since the Md 5.0 earthquake

(the epicenter is 120.4023°W, 36.2245°N) that occurred in the research area on July 25, 1983, we have taken the October 25,

1983 as the starting time. It is assumed that the stress in the crustal increases uniformly with time, we divide this period of

time into ten equal parts as we do in the experiments. We make this assumption because the interseismic slip velocity in this

region is almost stable and the mean occurrence time of Mw 6.0 earthquakes is about 20 years (Barbot et al., 2012), which is

consistent with the time interval that we choose. In order to better display the seismic activity of the research area and

surrounding areas in the corresponding period, we plot evolution maps of epicenter of a large area (116-122°W, 32-37°N)

that including the research area, as shown in Fig. 5. We select all earthquakes in this large area with magnitudes greater than

2.5 and occurrence time within the above range, with a total of 23,648 records.

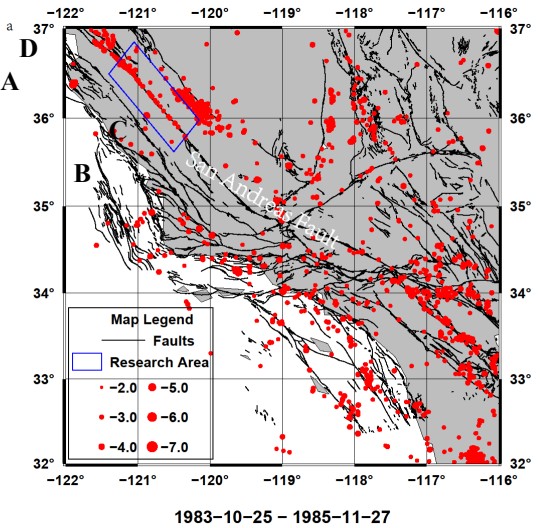

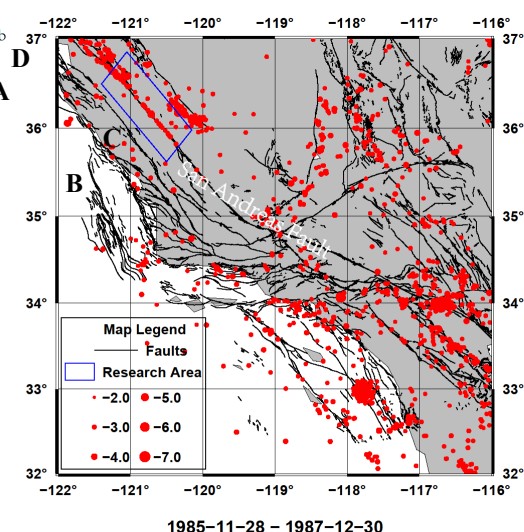






1987−12−31 − 1990−02−02

1990−02−03 − 1992−03−05

1992−03−06 − 1994−04−08

1994−04−09 − 1996−05−11

**Figure 5.** The seismicity of the research area and surrounding area in a seismic cycle

The red points indicate the epicenter of the earthquakes in the corresponding time. The location of the San Andreas Fault and the epicenter of the Parkfield earthquake are indicated on the map. The corresponding time is shown under each image.

(a) Seismicity of the research area and surrounding area during October 25, 1983 to November 27, 1985. (b) Seismicity of the research area and surrounding area during November 28, 1985 to December 30, 1987. (c) Seismicity of the research area and surrounding area during December 31, 1987 to February 02, 1990. (d) Seismicity of the research area and surrounding area during February 03, 1990 to March 05, 1992. (e) Seismicity of the research area and surrounding area during March 06, 1992 to April 08, 1994. (f) Seismicity of the

research area and surrounding area during April 09, 1994 to May 11, 1996. (g) Seismicity of the research area and surrounding area during May 12, 1996 to June 14, 1998. (h) Seismicity of the research area and surrounding area during June 15, 1998 to July 17, 2000. (i) Seismicity of the research area and surrounding area during July 18, 2000 to August 20, 2002. (j) Seismicity of the research area and

surrounding area during August 21, 2002 to September 28, 2004.

At each stage, there are many earthquakes with magnitude less than 5 in the study area, and many of them distribute along the fault in the research area. Qualitatively, this is consistent with the experimental results. Quantitatively, if these small earthquakes are compared with the sampling points with large deformation on the samples in the experiments, whether the

average distance between these small earthquakes and the fault zone is consistent with the experimental results will be essential.

### 2.5 Seismic monitoring models

The CV results calculated by the differential maximum principal strain in the experiments show that the marble samples with prefabricated cracks have obvious precursor characteristics before rupture, so it is essential to arrange the seismic monitoring

stations to capture such features. Three seismic monitoring models are proposed here: 1. Seismic stations are uniformly distributed in the study area; 2. Seismic stations are densely distributed along the fault zone, and the further away from the fault zone, the more sparse the distribution of the stations along the parallel direction of the fault zone is and the spacing of the stations along the vertical direction of the fault zone remains unchanged; 3. Seismic stations are densely distributed in the vertical direction of the fault zone, the further away from the fault zone, the more sparse the distribution of the stations in the

vertical direction of the fault zone is and the spacing of the stations in the parallel direction of the fault zone remains unchanged. The designs of the models are shown in Fig. 6 (sample 4 is taken as an example, and other samples are shown in the Appendix C).

Using the differential maximum principal strain as the monitoring signal, simulate with these three models when the numbers of seismic stations are 289, 196, and 100 respectively (as shown in Fig. 6). By comparing the CV obtained from

monitoring all sampling points and limited sampling points of different models, which model is more suitable for guiding the distribution of seismic stations and monitoring the precursory features of fracture and earthquakes is determined.

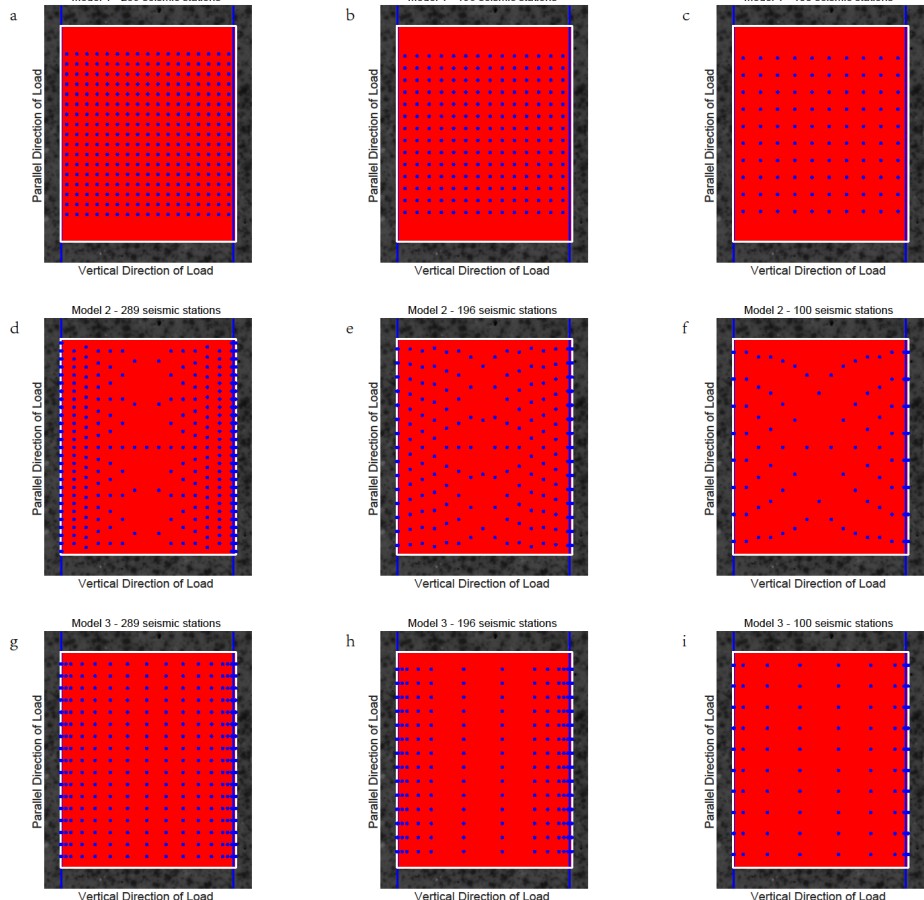

**Figure 6.** Three kinds of seismic monitoring models with different numbers of seismic monitoring stations

Each figure shows the observational area of sample 4. The blue lines indicate the prefabricated cracks. The area enclosed by the white

rectangle is the calculation area. The red points in the calculation area represent all the sampling points while the blue points indicate the

limited sampling points (seismic monitoring station) in different models. The horizontal axis is perpendicular to the direction of load and

the vertical axis is parallel to the direction of loading.

(a) Model 1 with 289 seismic stations. (b) Model 1 with 196 seismic stations. (c) Model 1 with 100 seismic stations. (d) Model 2 with 289

seismic stations. (e) Model 2 with 196 seismic stations. (f) Model 2 with 100 seismic stations. (g) Model 3 with 289 seismic stations. (h)

Model 3 with 196 seismic stations. (i) Model 3 with 100 seismic stations.

## 3. Results

One of the starting points of this paper is to explore whether the marble rocks with prefabricated cracks have precursors before fracture. To investigate this, different kinds of CV are calculated with different physical quantities obtained by DSCM,

and the fluctuation of each CV with the increase of load is observed. Our results show that each CV is fluctuating with an obvious jump in the loading process. Thus, we have selected the differential maximum principal strain with the most obvious characteristics as an example to show this. Besides, the locations of sampling points with large differential maximum principal strain exceeding the threshold at different loading stages are shown in Fig. 4 and Appendix B. The positions of such points change with the increase of load and gradually move towards the precast cracks. We compare this feature with

the seismicity in and around the northern end of the SAF in California, in order to establish a connection between experimental observation and natural observation. Finally, the differential maximum principal strain is taken as the monitoring signal to explore how to use limited stations to monitor the precursor characteristics more effectively.

## 3.1 Precursor characteristics

There are many physical quantities obtained by DSCM, including displacement and strain. We have used proposed method in Sect 2.3 to gain the corresponding differential values and compute the CV of these differential values, which are shown in Fig. 3. It can be seen obviously from Fig. 3 (and Appendix A) that each CV curve shows at least one jump during loading, which is the so-called precursor characteristics. Here, the CV calculated by the differential maximum principal strain is taken as an example to illustrate these precursor characteristics, because it has physical significance and obvious consequence.


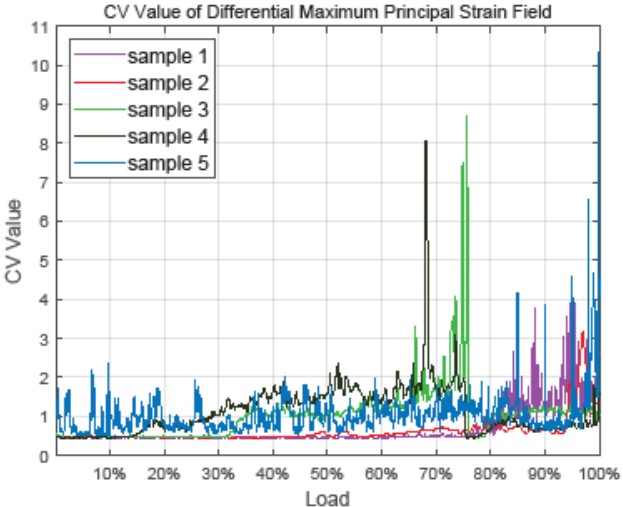

**Figure 7.** The CV of differential maximum principal strain field for the five samples

The colored lines represent the CV calculated by differential maximum principal strain field of different samples.

The CV of these five samples have a common background value (about 0.5) shown in Fig. 7, which proves the CV is a

statistic that can be used to describe the characteristics of different samples. Besides, each CV curve fluctuates with the

increase of load and some of them reach the largest level at 60-80% fracture load, while others reach at 80-100%. We believe

that the reason for this difference is the uniqueness of each sample, which includes micro-cracks, porosity, joints, and so on.

During direct shear, joints dilate before slip (Goodman, 1970; 1973), and even after many stick-slip cycles a small amount of

dilation is observed before each event (Sundaran, 1976). Therefore, it is reasonable to believe that cracks begin to form in

samples when the CV reaches high level, and local deformation is also relatively large and concentrated at this time. When

the cracks are connected, the whole will sample ruptures. Obviously, the CV indicates the development of cracks in the

samples and it is sensitive to deformation within the rocks.

### 3.2 Comparison with nature seismicity

By observing the CV of different samples, we set different thresholds for these samples and mark the sampling points with

large differential maximum principal strain that exceed the corresponding threshold on each sample's surface at different





loading stages (Fig. 4 and Appendix B). It can be seen clearly from these results, the positions of these sampling points may

be disordered at the beginning. This disorder is not truly disordered, but is actually affected by the development of cracks

and stress concentrations inside the rock during loading. When the rock is close to the rupture stage, these large deformation

points appear around the precast cracks. Therefore, it is necessary to investigate the changes of the average distance between

these points and the precast cracks as the load increases if we want to know whether the locations where these points appear

are regular. We calculate the distance between these points and the right precast crack of each sample, since most of these

points are concentrated on the right crack at the final stage. Furthermore, understanding the relationship between the

locations of small earthquakes (magnitude < 5) and faults, before moderate and strong earthquakes (magnitude ≥ 5), is also

the key to connect the experimental results with the natural observations. Thus, the experimental results are compared with

the seismicity of the northern end of the SAF and its surrounding area (the research area), and whether these two have

similar characteristics is analyzed.

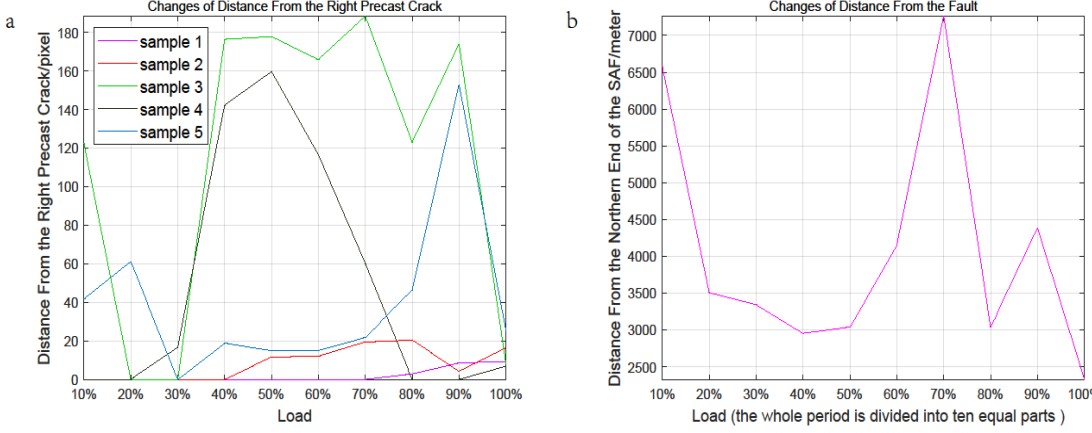

**Figure 8.** Changes of distance

(a) The average distance between the positions of the sampling points with large differential maximum principal strain and the right

precast crack. The colored lines represent the results of different samples. (b) The average distance between the small earthquakes

(magnitude < 5) and the northern end of SAF in the research area.

when there is no sampling point with large differential maximum principal strain satisfying the judge condition (we set for

each sample in chapter 2.4) in the corresponding loading stage, we assumed that the average distance from these sampling

points to the right precast crack in this stage is zero. The results obtained under this assumption are shown in Fig. 8. It can be seen from the Fig. 8 that the average distance of most samples and natural earthquakes are relatively stable at 40-60% loading stage, and then the distance changes dramatically as the load increases. For some samples (sample 3 and sample 5), there is a peak at 90% loading stage, which is consistent with the actual result in Fig. 8b. Finally, as the fracture approaches,

the average distance between the sampling points with large differential maximum principal strain of all the samples and the right prefabricated crack become small, which proves that these sampling points are clustered around the crack at last. It is worth noting that the results of all samples are close to the same value at 100% loading stage, which also indicates that the sampling points meeting the judge condition are concentrated around the right prefabricated crack and the concentration degree is nearly the same. The actual result in Fig. 8b shows that the positions of small earthquakes converge towards the

fault with the approaching of moderate and strong earthquakes, which is also the same as the experimental results. In particular, the result of sample 5 is showing striking similarity with the actual result. A very important phenomenon here is that the maximum value of these distances occur at a certain stage rather than the initial stage of loading. This means that a moderate or strong earthquake near the fault is possible soon after many small earthquakes have occurred in places far from the fault. This will be helpful to understand the development of ground strain and spatial evolution characteristics of

earthquakes.

### 3.3 Comparison of seismic monitoring models

After showing the precursor characteristics of fracture, how to monitor this kind of precursory signal effectively becomes quite significant. Three commonly used seismic monitoring models are presented in Sect 2.5. In fact, these three models are

equivalent to taking a limited number of sampling points in the calculation area of sample surface in three different ways. Here, the differential maximum principal strain of the chosen sampling points is used as the monitoring signal to compare the monitoring effects of these models. There are three steps to achieve this aim: firstly, calculate the CV of the limited sampling points monitored by the three models; secondly, calculate the CV of the full sampling points; finally, calculate the correlation coefficient of these two CV with the following formula.






$$r = \frac{\sum_{i=1}^{n}(x_i-\overline{x})(y_i-\overline{y})}{\sqrt{\sum_{i=1}^{n}(x_i-\overline{x})^2 \sum_{i=1}^{n}(y_i-\overline{y})^2}} \qquad (5)$$

In this formula, $\overline{x}$ is the mean differential maximum principal value of the limited sampling points ($\{x_1, x_2, ...x_n\}$),

and $\overline{y}$ is the mean differential maximum principal value of the full sampling points ($\{y_1, y_2, ...y_n\}$). The results are shown

in Table. 2.


**Table 2.** Monitoring results of different models

| Correlation coefficient / Samples — Models with different number of seismic stations | | Sample 1 | Sample 2 | Sample 3 | Sample 4 | Sample 5 |
|---|---|---|---|---|---|---|
| Model 1 | 289 | 0.9631 | 0.9210 | 0.9691 | 0.9692 | 0.9806 |
|  | 196 | 0.9515 | 0.9290 | 0.9383 | 0.9622 | 0.9119 |
|  | 100 | 0.7575 | 0.8320 | 0.7016 | 0.7991 | 0.9147 |
| Model 2 | 289 | 0.6946 | 0.9731 | 0.9572 | 0.7534 | 0.9803 |
|  | 196 | 0.9534 | 0.9382 | 0.9491 | 0.7425 | 0.9834 |
|  | 100 | 0.5403 | 0.8700 | 0.7314 | 0.5377 | 0.9252 |
| Model 3 | 289 | 0.8185 | 0.9601 | 0.9313 | 0.7609 | 0.9806 |
|  | 196 | 0.6819 | 0.9115 | 0.9165 | 0.7443 | 0.9622 |
|  | 100 | 0.5304 | 0.8588 | 0.7684 | 0.6601 | 0.9609 |

Generally, when the number of seismic stations is sufficient (≥289), arranging stations uniformly like Model 1 throughout the research area is the best way for monitoring precursory signal because most of the corresponding correlation coefficients are giving highest values. When the number of seismic stations is relatively small (196 seismic stations), the monitoring

effect of Model 2 is the best, even better than that of Model 1. When the number of seismic stations continues to decrease (less than 100), the monitoring effects of the three models have almost no difference. Therefore, these three models are suitable for different situations. In the case of a small number of seismic stations, selecting any model is fine due to the similar monitoring effect. As the number of stations increases, the advantages of Model 2 and Model 1 begin to emerge. According to this study, a more appropriate way to arrange seismic stations can be chosen in field work.

**4. Discussion**

**4.1 The CV fluctuates with the increase of load**

It is apparent that the CV of different samples do fluctuate with the increase of load and appear dramatically jumps. Simultaneously, some of these jumps occur at the 60-80% loading phase while the others happen at the 80-100% loading stage. Although the existence of premonitory characteristics is proved by these jumps, the curve features of each CV are not

the same. Each one has its own maximum value and fluctuation characteristics, which is why we set various thresholds to find large deformation sampling points on different sample surfaces. The causes of the distinctions are worth pondering. The main account for these dissimilarities is the inherent properties of the samples, including porosity, connectivity, and degree of joint development. As the crack density increases the crack interactions become more significant (Sieradzki and Li, 1986), which is responsible for the rupture. Thus, the first obvious CV jumps occurring during diverse loading phases of the curves

for these samples reveal the initial inherent level of crack density inside the samples when other properties are the same for all the samples. It is not only porosity that affects the characteristics of the CV curves, but also connectivity and degree of joint development. The higher they are, the smaller the maximum value of the CV will be, vice versa.

Another special phenomenon of CV curves is that each curve leaps significantly during a specific period of the whole loading stage. Some of them reach a high level at the 60-80 percent loading stage while the others jump at the 80-100

percent loading stage. If we regard the samples fracture as a main earthquake, then this kind of jump is a concentrated

release of stress, resulting in the emergence of large deformation sampling points that can be considered as some small earthquakes before the main shock. The occurrence time of the CV jumps suggests that these small earthquakes can be triggered much earlier or just a little earlier than the main shock. When the former occurs, there will be a period of quiescence before the main earthquake, and if the latter happens, there will be an increase in seismicity before the main

shock. The law of seismicity shows that before the occurrence of a large earthquake, the small earthquake activity may increase rapidly or decrease or even calm (Wyss, 1997) in the near epicenter area of the large earthquake. There is controversy about these two. Some researchers find that there may be an acceleration period of seismic activity rather than the quiescence before some violent earthquakes (Bowman and King, 2001; Chen, 2003). However, with the development of seismic monitoring methods and the improvement of earthquake catalogue, more and more phenomena

of quiescence before large earthquakes have been found (Wu and Chiao, 2006; Katsumata, 2011; Pu, 2018). The relationship between accelerating seismicity and quiescence is also highly regarded, as they are two major phenomena ahead of a main shock (Di Giovambattista and Tyupkin, 2004; Mignan and Di Giovambattista, 2008). The results of CV in this paper are also showing these two precursory characteristics, which indicates that this statistic is effective in describing and extracting premonitory features of rock fracture.

**4.2 Connection between experimental results and natural seismicity**

In fact, our experiments is aim to simplify the complex mechanism of natural seismicity around a strike-slip fault and simulate the deformation process in one seismic cycle. However, the scale and model problems need to be considered during this procedure, in which the scale problem refers to the conversion between laboratory scale and natural scale and the model question refers to whether the laboratory model can be used in nature. Therefore, we set the rectangular area

with only one strike-slip fault shown in Fig. 5 as a study area to limit the influence of other faults around, so that the simulations of the experiments are consistent with the natural state in a certain extent, which help us to weaken the above two problems.

There are many similarities between some experimental results and actual results, including an increase in distance at 90% loading stage and a decrease at 100% loading stage, as shown in Fig. 8. However, some samples do not show such a

remarkable increase and decrease and the reason for this difference can be divided into two parts. The first part is relevant

to intrinsic properties of the samples as detailed in Sect 4.1, which is also the main reason for the causes of this

dissimilarity. The second part may be related to the practical dimensions of the samples and the prefabricated cracks on

these samples, but this part may have just a little influence because the errors are very small and within the permit. A

convincing argument for observation is that the final results of all samples are very close, which indicates that the sampling

points with large deformation gather around the precast crack at this time. Besides, some small earthquakes may occur a

little farther from the fault before the main earthquake and this is the meaning of the sudden jump at 90% loading stage

shown in Fig. 8. Investigating whether other areas with a relatively stable seismic cycle and a strike-slip fault also have

these features is a useful way to understand the seismogenic mechanism and the seismicity included in this phenomenon.

### 4.3 effects of earthquake monitoring models

After finding that the CV is effective to characterize the precursor of rock fracture and the experimental results have some

common features with the natural seismicity results, we design three monitoring models to explore how to capture it more

effectively. Under the premise that each sampling point is regarded as a seismic station, these three models are actually

equivalent to extracting corresponding sampling points in three ways for monitoring. The monitoring effect is judged by the

correlation coefficient between the CV calculated by differential maximum principal strain of limited sampling points and all

the sampling points of the models. We set a different number of stations for each model to compare the monitoring effect of

them comprehensively. There is no doubt that distributing station evenly like Model 1 is the best way to seize these

precursors while the number of stations is sufficient. But if the number of stations is dwindled, unexpected results emerge.

The monitoring effect of Model 1 is no longer occupies a dominant position, and the effect of the distribution mode of Model

2 has surpassed that of Model 1. If the number of stations continues to decrease, the monitoring effect of the three models

showed a little difference. Starting from the formula of the correlation coefficient, the high value can be achieved if the CV

of the three models coincides with that of the full sampling points. Thus, the monitoring model can have a relatively high

correlation coefficient as long as it can ensure that the ratio of the sampling points with large deformation and the sampling

points with small deformation conforms to that of the full sampling points. In this case, arranging the limited seismic stations
uniformly in the entire area is the best way to capture the precursors if the numbers of seismic stations are large enough.

However, the number of stations is very small at present, so it is beneficial for field work to explore how to distribute stations. This is a simulation of the earthquake monitoring model with a limited number of stations, hoping to provide a little help for related work.

## 5. Conclusion

Experiment is an effective tool to understand the complex mechanism of natural earthquakes. We perform uniaxial loading

on five marble samples with prefabricated cracks and obtain their differential displacement and strain fields at different loading stages. The CV obtained from the calculation of these fields confirms the existence of precursor characteristics before rock fracture. Using results of the CV to set different thresholds, we find that large deformed sampling points on each sample surface will migrate to prefabricated cracks when the sample is close to failure. Similar features have been found on the seismicity of the San Andreas Fault in the research area. This is an attempt to link the experiment with the nature. All

these results prove the validity of the CV and the credibility of the CV describing the precursors. Thus, in order to monitor the precursory characteristics of this kind of rupture more effective, we have designed three commonly useable seismic monitoring models, and compare the monitoring effects of these models under the condition of limited seismic stations. It is found that the results of Model 1 and 2 are generally better than Model 3. In the field work, the most proper arrangement of seismic stations shall be selected according to the conditions, including the number of stations and the geological situation.


Code and Data availability. We shared the code and analyzed all the data of samples which available at: https:// disk.pku.edu.cn:443/link/FAEEA32750C2EAE8DC673C9AC2336267.

Author contributions. Andong Xu and Yonghong Zhao contributed to the conception of the study. Andong Xu performed all

the experiments with Jiaying Yang, Qi Zhang and Ru Liu. Andong Xu also performed the data analyses and wrote the manuscript. Muhammad Irfan Ehsan helped revising the manuscript.



Competing interests. The authors declare that they have no conflict of interest.

Acknowledgements. We thank all the people who are helpful to this article. This work was funded by National Key Research

and Development Program of China (Grant Number is 2018YFC1504203, SQ2017YFSF040025).





# Appendices

## Appendix A. CV images of the other samples

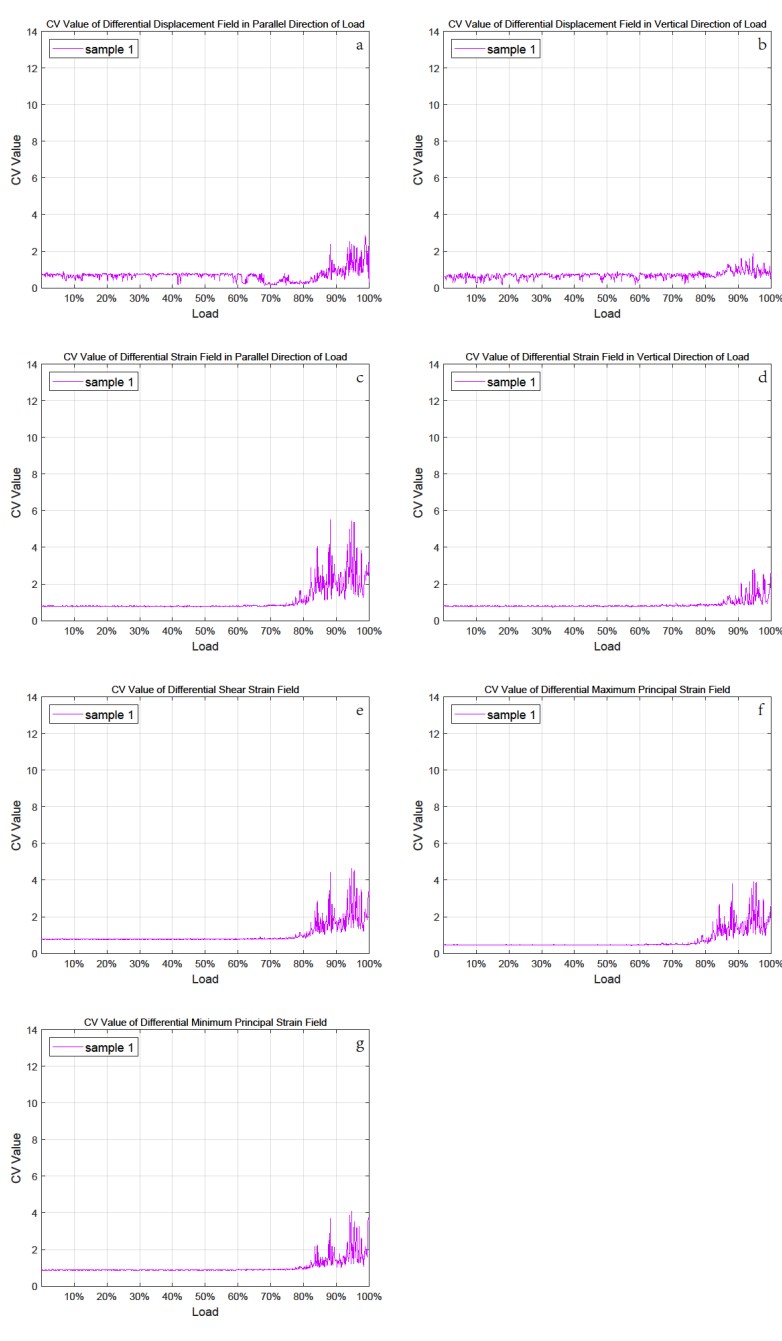


**Figure A1**. CV images of sample 1

(a) The image of CV changing with load, which is obtained from the differential displacement field in parallel direction of load. (b)The image of CV changing with load, which is obtained from the differential displacement field in vertical direction of load. (c) The image of CV changing with load, which is obtained from the differential strain field in parallel direction of

load. (d)The image of CV changing with load, which is obtained from the differential strain field in vertical direction of load. (e) The image of CV changing with load, which is obtained from the differential shear strain field. (f) The image of CV changing with load, which is obtained from the differential maximum principal field. (g) The image of CV changing with load, which is obtained from the differential minimum principal strain field.





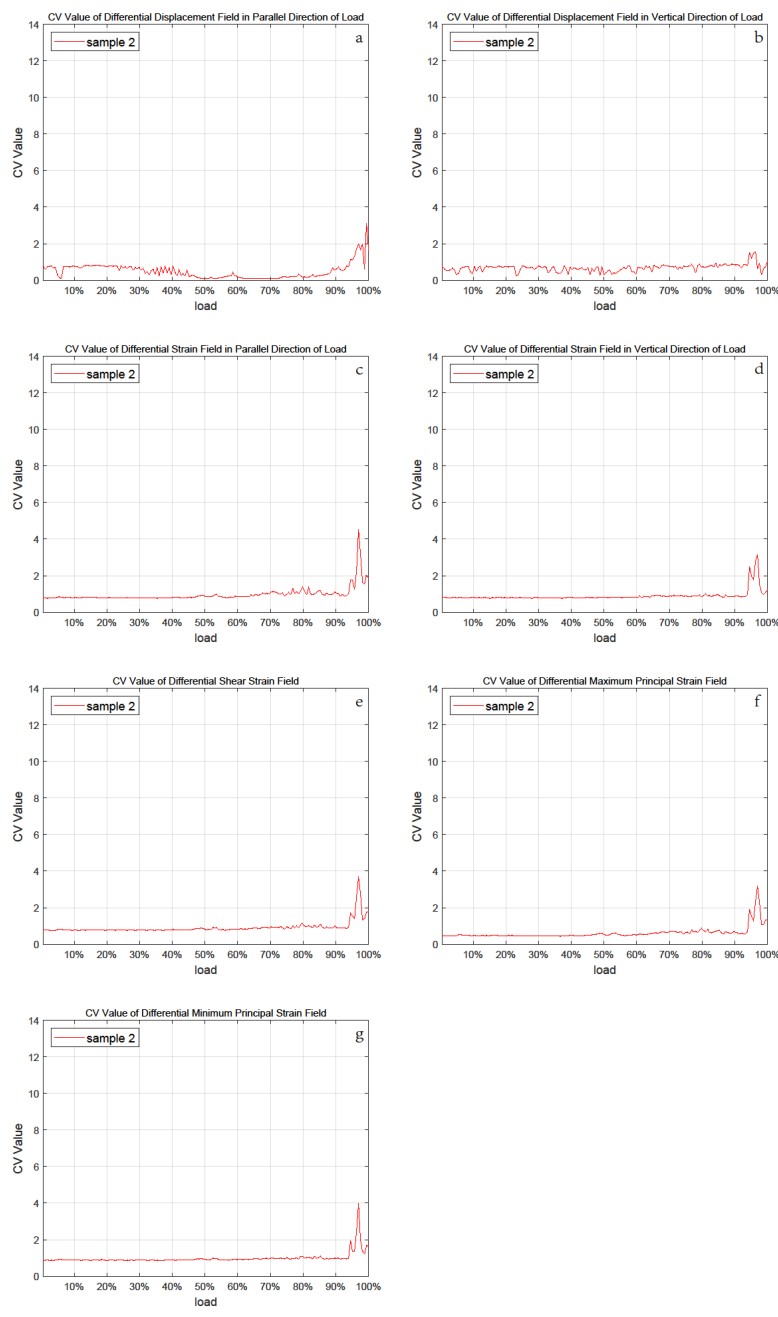

**Figure A2**. CV images of sample 2

(a) The image of CV changing with load, which is obtained from the differential displacement field in parallel direction of

load. (b)The image of CV changing with load, which is obtained from the differential displacement field in vertical direction





of load. (c) The image of CV changing with load, which is obtained from the differential strain field in parallel direction of load. (d)The image of CV changing with load, which is obtained from the differential strain field in vertical direction of load. (e) The image of CV changing with load, which is obtained from the differential shear strain field. (f) The image of CV changing with load, which is obtained from the differential maximum principal field. (g) The image of CV changing with

load, which is obtained from the differential minimum principal strain field.

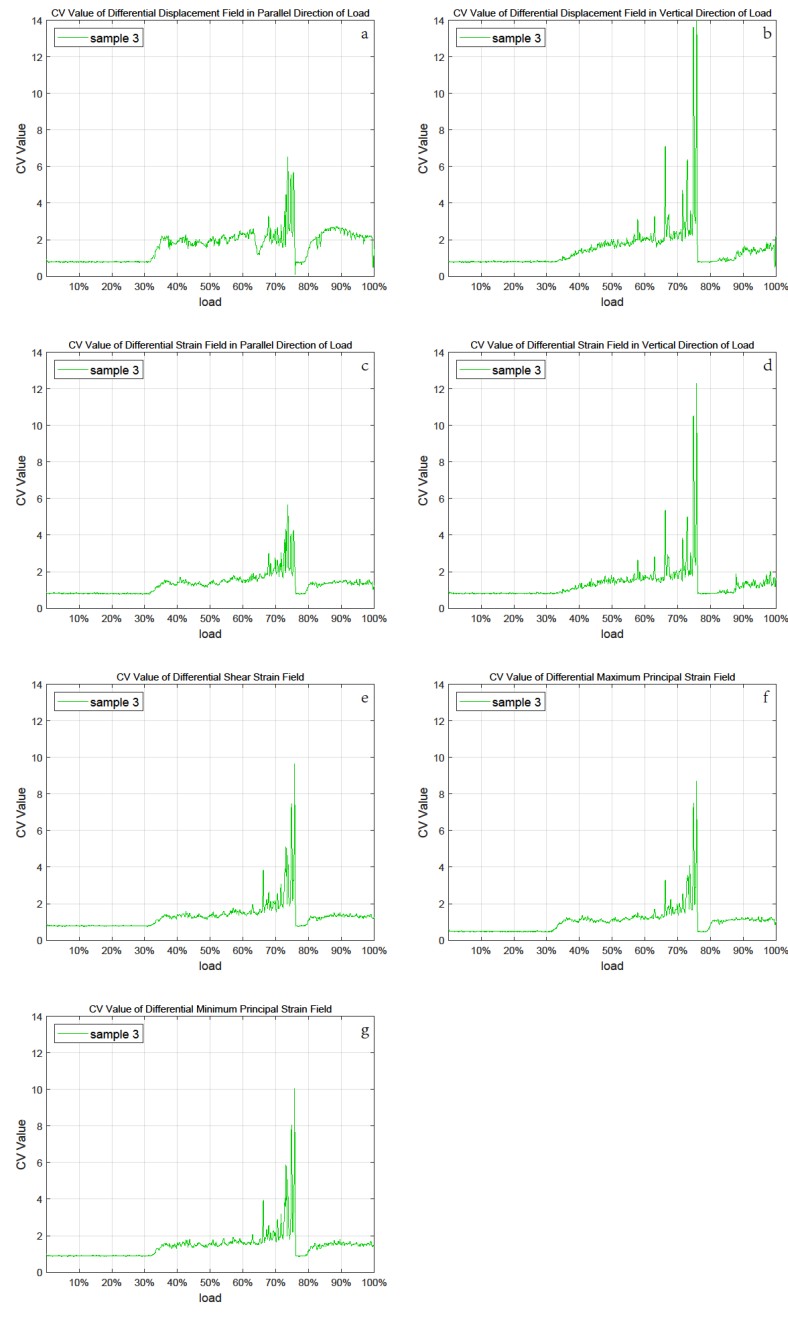

**Figure A3**. CV images of sample 3

(a) The image of CV changing with load, which is obtained from the differential displacement field in parallel direction of

load. (b)The image of CV changing with load, which is obtained from the differential displacement field in vertical direction





of load. (c) The image of CV changing with load, which is obtained from the differential strain field in parallel direction of load. (d)The image of CV changing with load, which is obtained from the differential strain field in vertical direction of load. (e) The image of CV changing with load, which is obtained from the differential shear strain field. (f) The image of CV changing with load, which is obtained from the differential maximum principal field. (g) The image of CV changing with load, which is obtained from the differential minimum principal strain field.

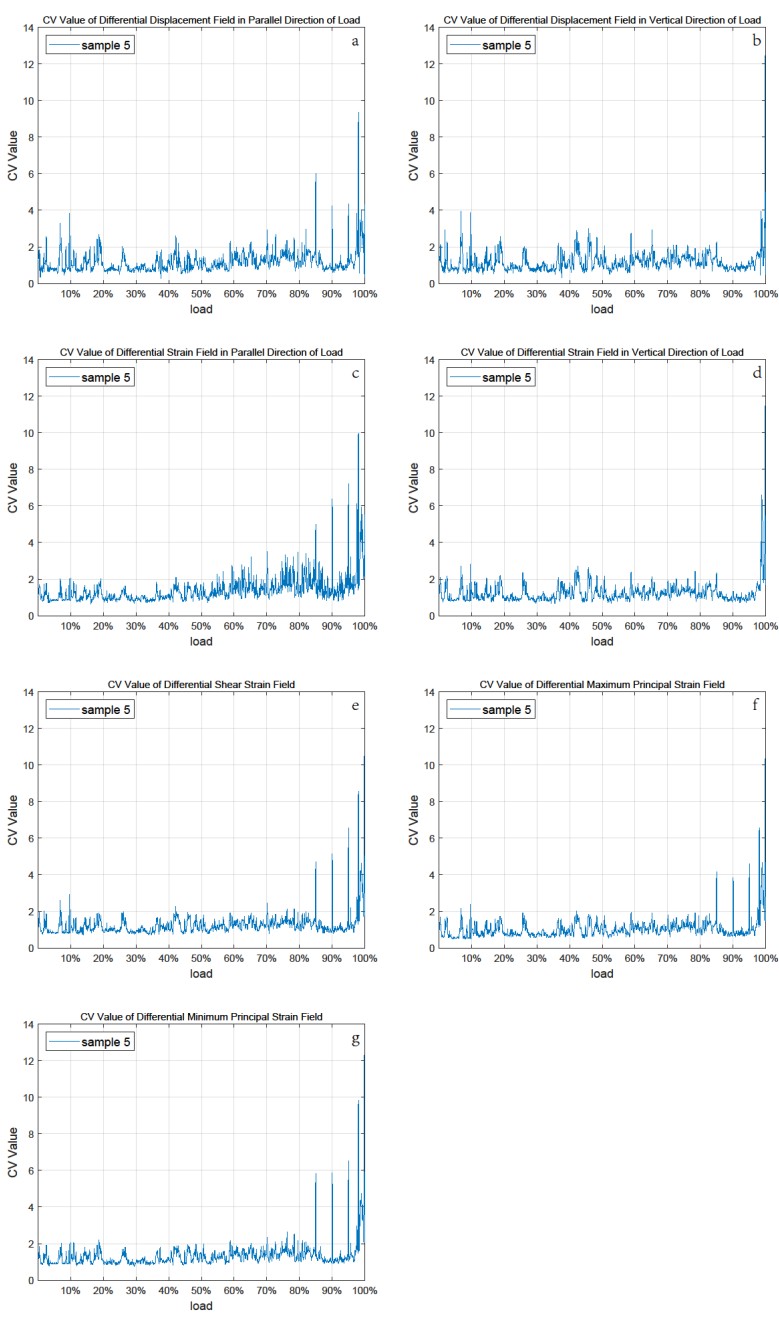

**Figure A4**. CV images of sample 5

(a) The image of CV changing with load, which is obtained from the differential displacement field in parallel direction of load. (b)The image of CV changing with load, which is obtained from the differential displacement field in vertical direction





of load. (c) The image of CV changing with load, which is obtained from the differential strain field in parallel direction of

load. (d)The image of CV changing with load, which is obtained from the differential strain field in vertical direction of load.

(e) The image of CV changing with load, which is obtained from the differential shear strain field. (f) The image of CV

changing with load, which is obtained from the differential maximum principal field. (g) The image of CV changing with

load, which is obtained from the differential minimum principal strain field.


**Appendix B. The position of the sampling points with large differential maximum principal strain for the other samples changes with load**



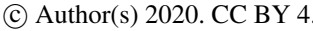





**Figure B1.** The position of the sampling points with large differential maximum principal strain for sample 1

The blue lines indicate the prefabricated cracks. Each figure shows the observational area of sample 4 in the experiments. The area enclosed by the white rectangle is the calculation domain and its size is constant in different loading stages. The red points represent the sampling points with large differential maximum principal strain that satisfy the judgment condition as the load increases.

(a) The result of the 10% loading stage. (b) The result of the 20% loading stage. (c) The result of the 30% loading stage. (d) The result of

the 40% loading stage. (e) The result of the 50% loading stage. (f) The result of the 60% loading stage. (g) The result of the 70% loading stage. (h) The result of the 80% loading stage. (i) The result of the 90% loading stage. (j) The result of the 100% loading stage.



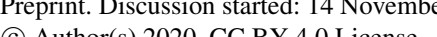





**Figure B2.** The position of the sampling points with large differential maximum principal strain for sample 2

520                                    All legends are consistent with **Figure B1.**





the Sampling Points with Large Differential Maximum Principal Strain in 10% load stage

the Sampling Points with Large Differential Maximum Principal Strain in 20% load stage

the Sampling Points with Large Differential Maximum Principal Strain in 30% load stage

the Sampling Points with Large Differential Maximum Principal Strain in 40% load stage

the Sampling Points with Large Differential Maximum Principal Strain in 50% load stage

the Sampling Points with Large Differential Maximum Principal Strain in 60% load stage

the Sampling Points with Large Differential Maximum Principal Strain in 70% load stage

the Sampling Points with Large Differential Maximum Principal Strain in 80% load stage

the Sampling Points with Large Differential Maximum Principal Strain in 90% load stage

the Sampling Points with Large Differential Maximum Principal Strain in 100% load stage



**Figure B3.** The position of the sampling points with large differential maximum principal strain for sample 3

All legends are consistent with **Figure B1.**




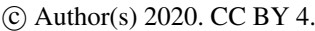



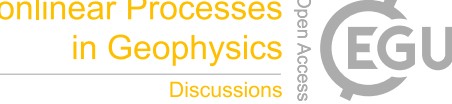

**Figure B4.** The position of the sampling points with large differential maximum principal strain for sample 5

All legends are consistent with **Figure B1.**

**Appendix C. Three kinds of seismic monitoring models with different numbers of seismic monitoring stations**

**Figure C1.** Three kinds of seismic monitoring models with different numbers of seismic monitoring stations

The blue lines indicate the prefabricated cracks. Each figure shows the observational area of sample 1. The area enclosed by the white

rectangle is the calculation area. The red points in the calculation area represent the sampling points. The blue points indicate the locations


of seismic stations in different models. The horizontal axis is perpendicular to the direction of load. The vertical axis is parallel to the

direction of loading.

(a) Model 1 with 289 seismic stations. (b) Model 1 with 196 seismic stations. (c) Model 1 with 100 seismic stations. (d) Model 2 with 289

seismic stations. (e) Model 2 with 196 seismic stations. (f) Model 2 with 100 seismic stations. (g) Model 3 with 289 seismic stations. (h)

Model 3 with 196 seismic stations. (i) Model 3 with 100 seismic stations.


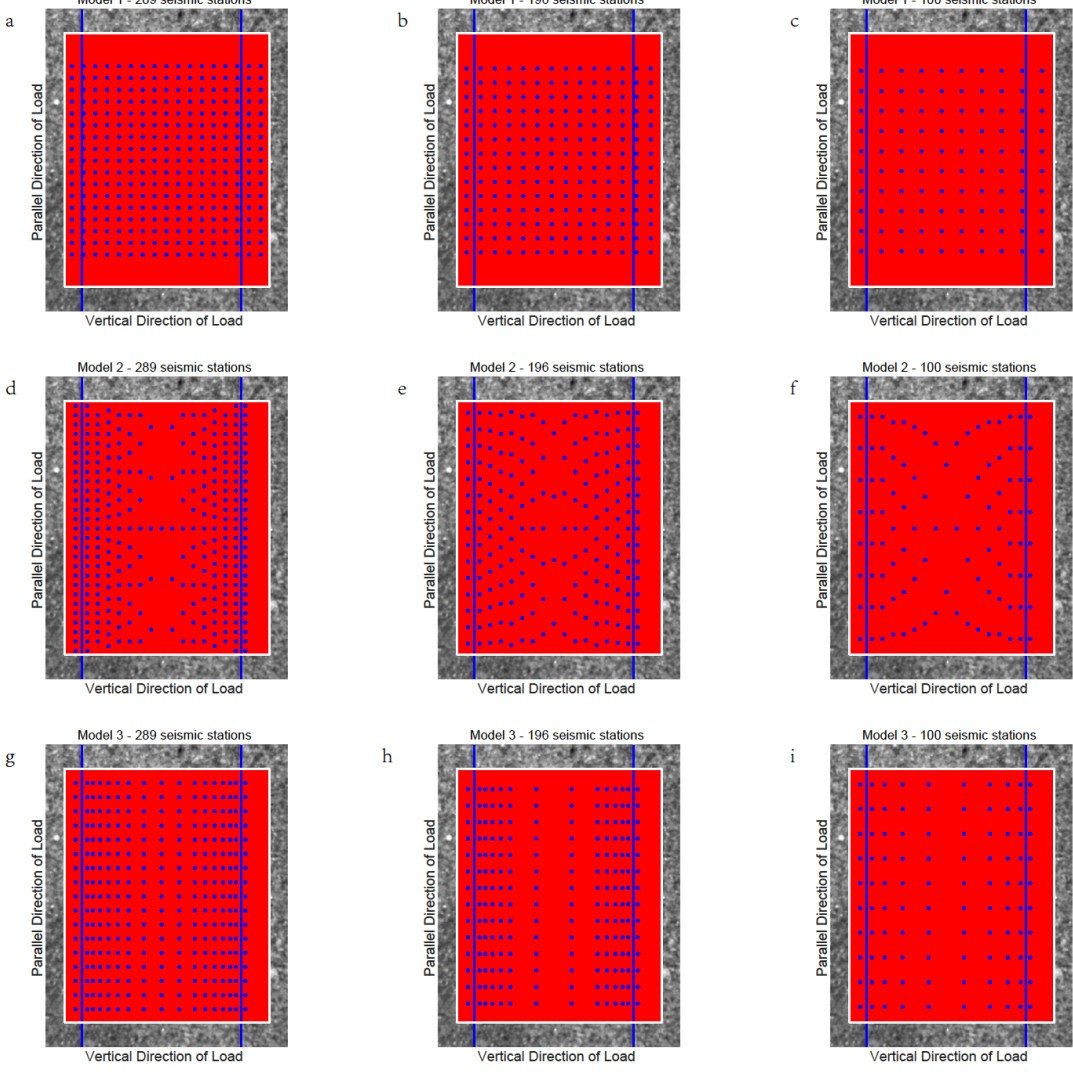

**Figure C2.** Three kinds of seismic monitoring models with different numbers of seismic monitoring stations




The blue lines indicate the prefabricated cracks. Each figure shows the observational area of sample 2. The area enclosed by the white

rectangle is the calculation area. The red points in the calculation area represent the sampling points. The blue points indicate the

locations of seismic stations in different models. The horizontal axis is perpendicular to the direction of load. The vertical axis is parallel

to the direction of loading.

(a) Model 1 with 289 seismic stations. (b) Model 1 with 196 seismic stations. (c) Model 1 with 100 seismic stations. (d) Model 2 with 289

seismic stations. (e) Model 2 with 196 seismic stations. (f) Model 2 with 100 seismic stations. (g) Model 3 with 289 seismic stations. (h)

Model 3 with 196 seismic stations. (i) Model 3 with 100 seismic stations.


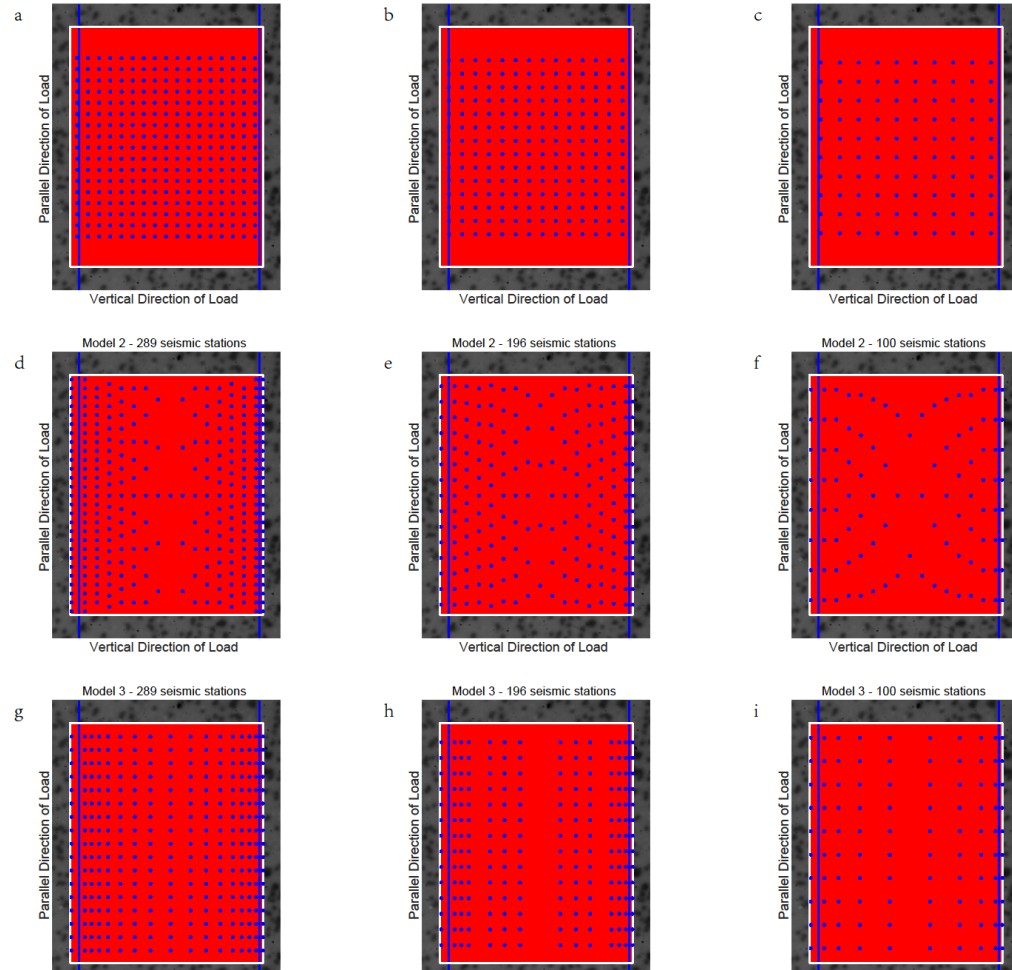



**Figure C3.** Three kinds of seismic monitoring models with different numbers of seismic monitoring stations

The blue lines indicate the prefabricated cracks. Each figure shows the observational area of sample 3. The area enclosed by the white rectangle is the calculation area. The red points in the calculation area represent the sampling points. The blue

points indicate the locations of seismic stations in different models. The horizontal axis is perpendicular to the direction of load. The vertical axis is parallel to the direction of loading.

(a) Model 1 with 289 seismic stations. (b) Model 1 with 196 seismic stations. (c) Model 1 with 100 seismic stations. (d) Model 2 with 289 seismic stations. (e) Model 2 with 196 seismic stations. (f) Model 2 with 100 seismic stations. (g) Model 3 with 289 seismic stations. (h) Model 3 with 196 seismic stations. (i) Model 3 with 100 seismic stations.





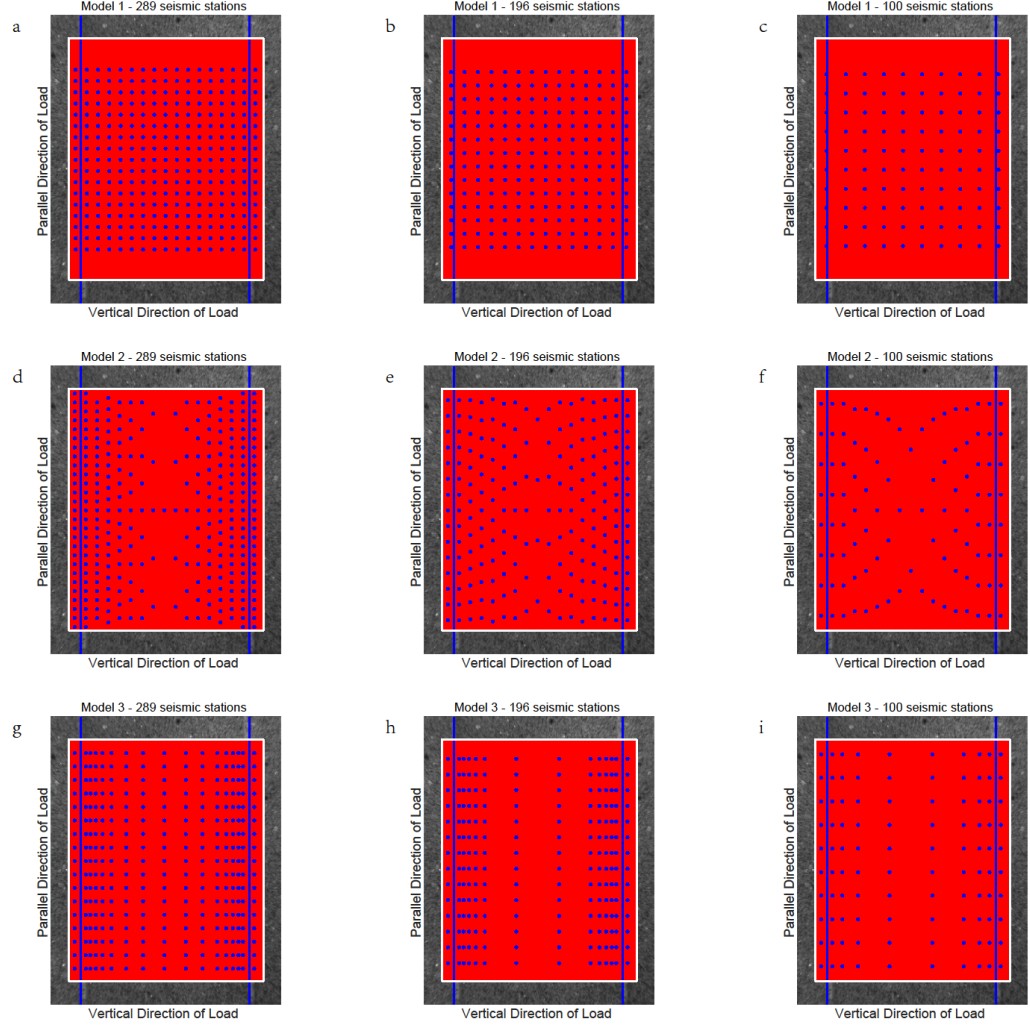

**Figure C4.** Three kinds of seismic monitoring models with different numbers of seismic monitoring stations

The blue lines indicate the prefabricated cracks. Each figure shows the observational area of sample 5. The area enclosed by

the white rectangle is the calculation area. The red points in the calculation area represent the sampling points. The blue

points indicate the locations of seismic stations in different models. The horizontal axis is perpendicular to the direction of

load. The vertical axis is parallel to the direction of loading.



(a) Model 1 with 289 seismic stations. (b) Model 1 with 196 seismic stations. (c) Model 1 with 100 seismic stations. (d)

Model 2 with 289 seismic stations. (e) Model 2 with 196 seismic stations. (f) Model 2 with 100 seismic stations. (g) Model 3

with 289 seismic stations. (h) Model 3 with 196 seismic stations. (i) Model 3 with 100 seismic stations.

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
