# Peer review of "Inhomogeneous precursor characteristics of rock with prefabricated cracks before fracture and its implication for earthquake monitoring"

_Nonlinear Processes in Geophysics, 2020_

## Referee Comment (RC1) · Anonymous Referee #1 · 6 Dec 2020

In this study, the precursory of rock fracture after special treatment is studied in laboratory, and compared with the actual scene, which has certain enlightenment significance for earthquake monitoring. The method of this study is reasonable and the conclusion seems reliable. There is only one suggestion: in the introduction and abstract, there is a lack of induction and generalization of the existing problems, which leads to the work of this paper. Due to the lack of such a summary, the two parts of the article seem to lack logic. Minor repairs are recommended.

---

## Referee Comment (RC2) · Anonymous Referee #2 · 29 Dec 2020

The investigation carried out by the authors is interesting in its attempt to identify shared features in laboratory and real-world seismicity. There a some minor but recurrent issues in grammar/vocabulary throughout the article that I believe need to be addressed. Presentation of some figures could be improved (some have unnecessary empty space and/or labels may be hard to read if printed).

Comments and questions: In sec 2.4 the authors choose magnitudes of 2.5 and above for the catalogue, but it is not clear why this particular choice was made and how this pertains to the completeness of the catalog in space and time, i.e., is the chosen SAF

catalog complete above this magnitude? (in other words, what is the magnitude of completeness for the chosen catalog and does changing these minimum magnitudes change the conclusions?) are there variations in time for this completeness specially after the larger events? Since the authors are speaking of links between experiment and 'natural' seismicity, it could be good to perhaps highlight/discuss the issues particular to each of the cases and where significant differences may lie between the two. For example; how would various types of incompletnesses (short-term aftershock incompleteness, catalog incompleteness etc.) affect their statements/conclusions? Within the context of the experimental setup, how are these incompletenesses accounted for? Given the brevity of the conclusions perhaps that section could be expanded to include some of these points along with a more elaborate synthesis of the statements in Sec.4.2. In Sec. 5, it could also be instructive and clearer to understand the overall message of the study by elaborating under which parameters/conditions the "... attempt to link the experiment with the nature" is made.

---

## Author Comment (AC2) · 31 Dec 2020

We would like to thank you for your positive comments, as well as the suggestions to improve the manuscript. We have checked the whole manuscript and revised the manuscript according to the referee (the new manuscript will be uploaded soon after contacting with the editor). The specific comments of the referee and our reply are as follows.

Comments:  There are some minor but recurrent issues in grammar/vocabulary

[Figure]

throughout the article that I believe need to be addressed. Presentation of some figures could be improved (some have unnecessary empty space and/or labels may be hard to read if printed).

Reply: We have revised some minor issues in our manuscript including expressions and figures, especially Figure 5 in which the labels is misaligned that due to the format convert of Word to PDF.

Comments and questions: In sec 2.4 the authors choose magnitudes of 2.5 and above for the catalogue, but it is not clear why this particular choice was made and how this pertains to the completeness of the catalog in space and time, i.e., is the chosen SAF catalog complete above this magnitude? (in other words, what is the magnitude of completeness for the chosen catalog and does changing these minimum magnitudes change the conclusions?) are there variations in time for this completeness specially after the larger events? Since the authors are speaking of links between experiment and 'natural' seismicity, it could be good to perhaps highlight/discuss the issues particular to each of the cases and where significant differences may lie between the two. For example; how would various types of incompletnesses (short-term aftershock incompleteness, catalog incompleteness etc.) affect their statements/conclusions? Within the context of the experimental setup, how are these incompletenesses accounted for? Given the brevity of the conclusions perhaps that section could be expanded to include some of these points along with a more elaborate synthesis of the statements in Sec.4.2. In Sec. 5, it could also be instructive and clearer to understand the overall message of the study by elaborating under which parameters/conditions the "... attempt to link the experiment with the nature" is made.

Reply: We choose magnitude of 2.5 and above for the catalogue because the chosen SAF catalog in one seismic cycle (1983-2004) above this magnitude is complete (Figure S1 in supplementary). We will add this statement in our manuscript due to the lack of clarification in the present manuscript. We analyze the catalogue completeness of magnitude of 2.5 and above, which shows slight variations with time divided in our

research and has less influence on our conclusion. However, we have not considered the variation in time for this completeness specially after the larger events yet, because the time we choose in this study is one seismic cycle that we think there is only one large event. The questions the referee proposed are very interesting and beneficial to our research in the future. In this study, we set different thresholds to capture the large deformation sampling points of different samples, corresponding to the magnitude of 2.5 and above for the catalogue we choose. How incompleteness corresponds to the experimental results in this study is not easy to analyzed, which is also not our target. We aim to link the experimental results with natural seismicity and hope to apply this foundation on earthquake monitoring.

Please also note the supplement to this comment:
https://npg.copernicus.org/preprints/npg-2020-44/npg-2020-44-AC2-supplement.pdf

———————————————————

[Figure]

**Fig. 1.**

**Supplement:**

[Figure]

**Figure S1.** Diagram of magnitude and frequency.

The relationship between the magnitude and frequency is close to linear, which follows the law of G-R. There is a small uplift when the magnitude larger than 6 that is normal in this area. Note that there exists an offset between the frequency of magnitude smaller than 2.5 and the linear value of fit, which may indicate an incompleteness of the seismic catalogue when the magnitude below 2.5. Therefore, we decide to choose magnitude of 2.5 and above for the completeness of catalogue.

---

## Author Response (AR1)

**Author's response to the referees**

1. Author's response to referee #1

(1) Comments from referee:

In this study, the precursory of rock fracture after special treatment is studied in laboratory, and compared with the actual scene, which has certain enlightenment significance for earthquake monitoring. The method of this study is reasonable and the conclusion seems reliable. There is only one suggestion: in the introduction and abstract, there is a lack of induction and generalization of the existing problems, which leads to the work of this paper. Due to the lack of such a summary, the two parts of the article seem to lack logic. Minor repairs are recommended.

(2) Author's response:

We would like to thank you for your positive comments, as well as the suggestions to improve the manuscript. Our starting point is to try to make a different type of analysis to link the laboratory observations with natural seismicity by comparing their similar characteristics. In the abstract, we emphasize the research contents and results of this paper. The induction and generalization of the existing problems, which leads us to do the work is explained in the introduction part. Firstly, we introduce that laboratory rock experiments are an effective method to study the properties of rocks and faults while directly measure these properties is extremely difficult. Secondly, we attempt to establish the relationship between laboratory observations and naturally seismicity, which is the existing and unsolved problem in this research field, by comparing the similar characteristics of them. Thirdly, we detailedly describe the theoretical basis, steps of our work and its implication for earthquake monitoring. We follow this logic, which we think is more reasonable and fluent.

(3) Author's changes in manuscript:

We have not revised our manuscript based on these comments because our aim is clear and the context what we write is logical.

2. Author's response to referee #2

We divided the comments of referee #2 into four parts because the number of comments is large and the context of them is detailed.

First part:

(1) Comments from referee:

The investigation carried out by the authors is interesting in its attempt to identify shared features in laboratory and real-world seismicity. There are some minor but recurrent issues in grammar/vocabulary throughout the article that I believe need to be addressed. Presentation of some figures could be improved (some have unnecessary empty space and/or labels may be hard to read if printed).

(2) Author's response:

We would like to thank you for your positive comments, as well as the suggestions to improve the manuscript. We have checked the whole manuscript and revised the manuscript according to the referee (the new manuscript will be uploaded soon after contacting with the editor). The specific comments of the referee and our reply are as follows.

(3) Author's changes in manuscript:

We have revised some minor issues in our manuscript including expressions and figures, especially Figure 5 in which the labels is misaligned that due to the format convert of Word to PDF. We revised 'which' to 'that' in line 93, and revised 'step' to steps in line 135. We added a space between number and unit, such as in line 151, 159 and so on. We also added a space between degree and direction in line 198, 199 and so on. Moreover, we changed some words and sentences in order to express more accurately, such as words in line 221, 222 and sentences in line 239. All of the revisions are highlighted in the new manuscript.

Second part:

(1) Comments from referee:

In sec 2.4 the authors choose magnitudes of 2.5 and above for the catalogue, but it is not clear why this particular choice was made and how this pertains to the completeness of the catalog in space and time, i.e., is the chosen SAF catalog complete above this magnitude? (in other words, what is the magnitude of completeness for the chosen catalog and does changing these minimum magnitudes change the conclusions?)

(2) Author's response:

We choose magnitude of 2.5 and above for the catalogue because the chosen SAF catalog in one seismic cycle (1983-2004) above this magnitude is complete (Figure S1 in supplementary). We will add this statement in our manuscript due to the lack of clarification in the present manuscript. We analyze the catalogue completeness of magnitude of 2.5 and above, which shows slight variations with time divided in our research and has less influence on our conclusion.

(3) Author's changes in manuscript:

We added the statement in our new manuscript in line 212–214 and analyzed the completeness of the catalog in Figure S1 in supplementary.

Third part:

(1) Comments from referee:

Are there variations in time for this completeness specially after the larger events? Since the authors are speaking of links between experiment and 'natural' seismicity, it could be good to perhaps highlight/discuss the issues particular to each of the cases and where significant differences may lie between the two.

(2) Author's response:

We have not considered the variation in time for this completeness specially after the larger events yet, because the time we choose in this study is one seismic cycle that we think there is only one large event.

(3) Author's changes in manuscript:

We have not added these discussions in our new manuscript because we choose only one seismic cycle. The variations in time for this completeness after the larger events may be included in next seismic cycle, which may change slightly because the seismic cycle in this region is relatively stable.

Four part:

(1) Comments from referee:

For example; how would various types of incompletnesses (short-term aftershock incompleteness, catalog incompleteness etc.) affect their statements/conclusions? Within the context of the experimental setup, how are these incompletenesses accounted for? Given the brevity of the conclusions perhaps that section could be expanded to include some of these points along with a more elaborate synthesis of the statements in Sec.4.2. In Sec. 5, it could also be instructive and clearer to understand the overall message of the study by elaborating under which parameters/conditions the "... attempt to link the experiment with the nature" is made.

(2) Author's response:

The questions the referee proposed are very interesting and beneficial to our research in the future. In this study, we set different thresholds to capture the large deformation sampling points of different samples, corresponding to the magnitude of 2.5 and above for the catalogue we choose. How incompleteness corresponds to the experimental results in this study is not easy to analyzed, which is also not our target. We aim to link the experimental results with natural seismicity and hope to apply this foundation on earthquake monitoring.

(3) Author's changes in manuscript:

We have not changed the context about this in our new manuscript because the incompleteness may do not exist in our research especially when we choose the magnitude of 2.5 and above for the catalogue. These questions are exactly what we are going to do next. However, the thresholds we set for different samples corresponds to magnitude of 2.5. In this case, we get the conclusions.

---

## Author Response (AR2)

**Author's response to the referees**

1. Author's response to referee #1

(1) Comments from referee:

In this study, the precursory of rock fracture after special treatment is studied in laboratory, and compared with the actual scene, which has certain enlightenment significance for earthquake monitoring. The method of this study is reasonable and the conclusion seems reliable. There is only one suggestion: in the introduction and abstract, there is a lack of induction and generalization of the existing problems, which leads to the work of this paper. Due to the lack of such a summary, the two parts of the article seem to lack logic. Minor repairs are recommended.

(2) Author's response:

Thank you for the summary and comments. They are very helpful in improving the quality of the manuscript. We tried to make a different type of analysis to link the laboratory observations with natural seismicity by comparing their similar characteristics, so that the existing problems were inducted and summarized less. We added this part in abstract and introduction, which is essential for the integrity of our research.

(3) Author's changes in manuscript:

We added this part in abstract and introduction, which is essential for the integrity of our research.

2. Author's response to referee #2

We divided the comments of referee #2 into four parts because the number of comments is large and the context of them is detailed.

First part:

(1) Comments from referee:

The investigation carried out by the authors is interesting in its attempt to identify shared features in laboratory and real-world seismicity. There are some minor but recurrent issues in grammar/vocabulary throughout the article that I believe need to be addressed. Presentation of some figures could be improved (some have unnecessary empty space and/or labels may be hard to read if printed).

(2) Author's response:

We would like to thank you for your positive comments, as well as the suggestions to improve the manuscript. We have checked the whole manuscript and revised the manuscript according to the referee. The specific comments of the referee and our reply are as follows.

(3) Author's changes in manuscript:

We have revised some minor issues in our manuscript including expressions and figures, especially Figure 5 in which the labels is misaligned that due to the format convert of Word to PDF. We revised 'which' to 'that' in line 93, and revised 'step' to steps in line 135. We added a space between number and unit, such as in line 151, 159 and so on. We also added a space between degree and direction in line 198, 199 and so on. Moreover, we changed some words and sentences in order to express more accurately, such as words in line 221, 222 and sentences in line 239. All of the revisions are highlighted in the new manuscript.

Second part:

(1) Comments from referee:

In sec 2.4 the authors choose magnitudes of 2.5 and above for the catalogue, but it is not clear why this particular choice was made and how this pertains to the completeness of the catalog in space and time, i.e., is the chosen SAF catalog complete above this magnitude? (in other

words, what is the magnitude of completeness for the chosen catalog and does changing these minimum magnitudes change the conclusions?)

(2) Author's response:

We choose magnitude of 2.5 and above for the catalogue because the chosen SAF catalog in one seismic cycle (1983-2004) above this magnitude is complete (Figure S1 in supplementary). We will add this statement in our manuscript due to the lack of clarification in the present manuscript. We analyze the catalogue completeness of magnitude of 2.5 and above, which shows slight variations with time divided in our research and has less influence on our conclusion.

(3) Author's changes in manuscript:

We added the statement in our new manuscript in line 212–214 and analyzed the completeness of the catalog in Figure S3 in supplementary.

Third part:

(1) Comments from referee:

Are there variations in time for this completeness specially after the larger events? Since the authors are speaking of links between experiment and 'natural' seismicity, it could be good to perhaps highlight/discuss the issues particular to each of the cases and where significant differences may lie between the two.

(2) Author's response:

We have not considered the variation in time for this completeness specially after the larger events yet, because the time we choose in this study is one seismic cycle that we think there is only one large event.

(3) Author's changes in manuscript:

We have not added these discussions in our new manuscript because we choose only one seismic cycle. The variations in time for this completeness after the larger events may be

included in next seismic cycle, which may change slightly because the seismic cycle in this region is relatively stable.

Four part:

(1) Comments from referee:

For example; how would various types of incompletnesses (short-term aftershock incompleteness, catalog incompleteness etc.) affect their statements/conclusions? Within the context of the experimental setup, how are these incompletenesses accounted for? Given the brevity of the conclusions perhaps that section could be expanded to include some of these points along with a more elaborate synthesis of the statements in Sec.4.2. In Sec. 5, it could also be instructive and clearer to understand the overall message of the study by elaborating under which parameters/conditions the "... attempt to link the experiment with the nature" is made.

(2) Author's response:

The questions the referee proposed are very interesting and beneficial to our research in the future. In this study, we set different thresholds to capture the large deformation sampling points of different samples, corresponding to the magnitude of 2.5 and above for the catalogue we choose. How incompleteness corresponds to the experimental results in this study is not easy to analyzed, which is also not our target. We aim to link the experimental results with natural seismicity and hope to apply this

foundation on earthquake monitoring.

(3) Author's changes in manuscript:

We have not changed the context about this in our new manuscript because the incompleteness may do not exist in our research especially when we choose the magnitude of 2.5 and above for the catalogue. These questions are exactly what we are going to do next. However, the thresholds we set for different samples corresponds to magnitude of 2.5. In this case, we get the conclusions.

3. Author's response to the editor

Comments from the editor:

The study performs statistical analysis of laboratory fracture experiment data to formulate a precursor based on the coefficient of variation of selected seismicity attributes. The methodology is then applied to the seismicity of the Parkfield segment of San Andreas Fault in California. The authors conclude that the results confirm "the existence of precursor characteristics before rock fracture" and "the credibility of the CV describing the precursors".

Author's response:

Reply: Thank you for the summary and comments. We tried to link experiments results with natural observations, which still has a long way to go.

Comments from the editor:

The general topic of the study -- testing selected labquake precursory phenomena in observed seismicity -- is interesting and appropriate for NPG. However, the science content and technical level of the work are below the standards of NPG. The work needs substantial restructuring and revising before it can be re-evaluated for possible publication.

Author's response:

Reply: Thank you for the summary and comments. They are very helpful in improving the quality of the manuscript. We agreed that the work needs substantial restructuring and revising and we uploaded a new version.

Comments from the editor:

I notice that the authors decided to either ignore or only cosmetically address the comments of the two reviewers -- this is unfortunate, since readers most probably will have similar concerns. I strongly suggest the authors to reconsider the reviewers' comments.

Author's response:

Reply: Thank you for reminding us. We addressed the comments of the two reviewers by

explaining how we thought and how we wrote this paper, which may be improper for answering the comments. We neglected the detailed answer for the specific problem and thus we reconsidered the two reviewer's comments.

Comments from the editor:

In addition, I notice the following issues:

1) The method description is unacceptable. It is very informal and sketchy, not allowing one to reproduce the results. This applies to the digital speckle correlation method, estimating displacement and strain from experiment observations, and defining the precursor in observations. The work in its current form cannot be evaluated, since the main examined quantities remain undefined.

Author's response:

Reply: Thank you for the comments. We revised the section of method with many specific descriptions, which is essential for evaluating the accuracy of this method. We also defined the examined quantities clearly in the revised manuscript.

Comments from the editor:

2) Please define the examined precursor (Is it increase of CV, decrease of the average distance to the crack, or something else?). A formal reproducible definition should be given. Next, I fail to see what is claimed to be a precursor in Fig. 8a. I see neither a systematic increase or decrease of the distance as a failure approach. With this variability and inconsistency of results in five lab samples, it is hard to expect that the precursor will work in observations. I also fail to see how Fig. 8b supports authors' conclusions. What do we see here -- an increase at 70% or decrease at 100%? What about stability of this result?

Author's response:

Reply: Thank you for the comments. We defined that the precursor is increase of CV, which occurs in all five samples. We aimed to say that CV can extract the precursor signal of rock samples before rupturing and it may be useful in natural research. Therefore, we used Fig. 8a

to investigate how the earthquakes distribute during a seismic cycle and compared it with the experimental results, which is an application of the CV. In Fig. 8a and 8b, there is a decrease near the 100 % loading stage and it reflects that foreshocks may cluster in an area near the active fault where mainshock occurs.

Comments from the editor:

3) Figures 3,4,5,6 have unnecessarily many panels with largely repeating information. Think of moving most of the panels to SI, and only keeping the most informative figures in the main text. The captions are repetitive -- please rewrite.

Author's response:

Reply: Thank you for the comments. We revised all of these figures and moved some of them into Supplementary. We also rewrote the repetitive captions.

Comments from the editor:

4) The text is clumsy, repetitive, and unnecessarily long. I suggest revising the presentation in order to formally show (with all relevant equations) and justify the main steps of the analysis. It is also advisable to improve the grammar and style.

Author's response:

Reply: Thank you for the comments. We removed the clumsy, repetitive and unnecessarily part in our paper as well as improved the grammar and style.

(3) Author's changes in manuscript:

We have revised all the content listed above and all the changes are highlighted by us.